# NK Cells in Cancer: Mechanisms of Dysfunction and Therapeutic Potential

**DOI:** 10.3390/ijms24119521

**Published:** 2023-05-30

**Authors:** Federica Portale, Diletta Di Mitri

**Affiliations:** 1Tumor Microenviroment Unit, IRCCS Humanitas Research Hospital, 20089 Milan, Italy; federica.portale@humanitasresearch.it; 2Department of Biomedical Sciences, Humanitas University, 20072 Milan, Italy

**Keywords:** NK cells, tumor microenvironment, immunotherapy

## Abstract

Natural killer cells (NK) are innate lymphocytes endowed with the ability to recognize and kill cancer cells. Consequently, adoptive transfer of autologous or allogeneic NK cells represents a novel opportunity in cancer treatment that is currently under clinical investigation. However, cancer renders NK cells dysfunctional, thus restraining the efficacy of cell therapies. Importantly, extensive effort has been employed to investigate the mechanisms that restrain NK cell anti-tumor function, and the results have offered forthcoming solutions to improve the efficiency of NK cell-based therapies. The present review will introduce the origin and features of NK cells, summarize the mechanisms of action and causes of dysfunction of NK cells in cancer, and frame NK cells in the tumoral microenvironment and in the context of immunotherapies. Finally, we will discuss therapeutic potential and current limitations of NK cell adoptive transfer in tumors.

## 1. Introduction

### 1.1. Natural Killer Cell Origin and Development

Natural killer (NK) cells are innate lymphocytes endowed with potent killing abilities against foreign, stressed, infected, and transformed cells that may be dangerous to the host. NK cells belong to the family of innate lymphoid cells (ILCs), a group of cells that participate in the innate immune response and thus play a key role in the early phase of host defense against infections and malignancies. The family of ILCs embraces five major groups, including natural killer cells, group 1 ILCs (ILC1s), group 2 ILCs (ILC2s), group 3 ILCs (ILC3s), and lymphoid tissue-inducer (LTi) cells. These subsets are defined by the expression of specific transcription factors that dictate cytotoxicity set against preferential targets and a specific cytokine release [1]. The site and timeline of origin of NK cells are still under debate. It is generally recognised that the early source of NK cells circulating in the blood is represented by CD34^high^ hematopoietic stem cells (HSCs), a subset of multipotent self-renewing cells that is enriched in the bone marrow (BM) in humans and mice [2,3]. Accordingly, the main developmental stages of NK cells, which include the differentiation of HSCs to common lymphoid progenitors (CLPs) towards NK cell progenitors (NKPs), have been described in the bone marrow. However, an extramedullary maturation and differentiation of NK cells has been reported to take place in secondary lymphoid Tissues (SLTs), such as the thymus and lymph nodes, which thus represent alternative sources of this cell subset in adults [4]. In general, from BM, NK cells migrate toward SLTs and peripheral organs [5,6,7]. During maturation, the expression of the CD56 (neural cell adhesion molecule (NCAM)) surface marker is indicative of the acquisition of distinct functional properties by human mature NK cells that become capable of self-tolerance as well as of recognition of pathogens and cellular alterations. CD56^bright^ NK cells are mostly cytokine releasers that acquire cytotoxic properties with further steps for maturation, designed by the downregulation of CD56 and the acquisition of type III Fcγ receptor CD16 and killer-cell immunoglobulin-like receptors (KIRs) [8,9]. Functionally similar NK cell subsets have been identified in mice, where regulatory and cytokine-producer NK cells are defined by the expression of tumor necrosis factor receptor superfamily member CD27 and the integrin CD11b. Upon maturation, murine NK cells lose the expression of CD27, and CD27^neg^CD11b^bright^ cells represent the terminally differentiated subset marked by strong effector capabilities [10,11]. Interestingly, mature NK cells are endowed with a potent cytotoxic activity that may be further augmented with the achievement of memory-like features that resemble those of adaptive lymphocytes. Memory-like responses have been observed in mature NK cells upon viral infection or following exposure to cytokines such as interleukin (IL)-12, IL-15, and IL-18, and memory-like NK cells (mlNK) show an enhanced activation once re-exposed to a stimulation. However, the acquisition of a memory phenotype by NK cells varies depending on the stimuli and can be diverse on the basis of location and disease [12,13,14,15].

### 1.2. Natural Killer Cell Migration to Peripheral Organs

Once mature, NK cells move from BM toward SLTs and peripheral non-lymphoid organs, including the liver, lungs, intestine, and uterus, and once there, undergo further steps of maturation and differentiation [5,6,7]. NK cell trafficking is modulated by a plethora of molecules, including integrins, selectins, and chemokines. During maturation, the progressive increase in the expression of the receptors Sphingosine-1-Phosphate 5 (S1P_5_) and C-X3-C Motif Chemokine Receptor 1 (CX3CR1), and the parallel downregulation of C-X-C Motif Chemokine Receptor 4 (CXCR4) determine the exit of NK cells from the BM sinusoids into the bloodstream, finally homing to secondary lymphoid organs and peripheral tissues [16]. Importantly, the trafficking of NK cells to definite organs is finely tuned by the interaction between specific receptors and ligands. For example, NK cell homing to LNs is mainly determined by the engagement of C-C Motif Chemokine Receptor 7 (CCR7) with C-C Motif Chemokine Ligand 19 (CCL19) and CCL21 [17].

As mentioned above, NK cells also exert their functions in non-lymphoid tissues. Among those, the liver and decidua are particularly infiltrated by NK cells that contribute to organ homeostasis. The liver shows a unique immunological environment, where both resident and recruited NK cells contribute to tolerance, in order to avoid chronic inflammation [18]. CXCR6 represents one of the key chemokine receptors involved in NK cell retention in the liver, due to its interaction with C-X-C Motif Chemokine Ligand 16 (CXCL16) released by the sinusoids [19], together with CCR1, CCR2, CCR3, CCR5, and CXCR3, and their respective ligands, including CCL2, CCL3, and CXCL10 [20]. In the decidua, NK cell abundance is essential to sustain pregnancy. Here, NK cell infiltration is mediated by the release of chemokines including CXCL10, CXCL12, CCL3, and C-X3-C Motif Chemokine Ligand 1 (CX3CL1), that induce the recruitment of NK cells from peripheral blood, mainly via the interaction with CXCR3 and CXCR4 [21]. As for the liver, NK cells are thought to contribute to lung homeostasis paralleling hepatic NK cell functions [22], and several lung-homing receptors, including CXCR3, CXCR6, CCR2, and CCR5, have been described [23]. Of interest, NK cells have been identified within the central nervous system (CNS), where they exert a key role in immunosurveillance. NK cells can enter the CNS through the blood–brain barrier and the choroid plexus, once recruited by chemokines such as CX3CL1, CCL2, and CXCL10 [24]. It has to be said that the modulation of the receptor–chemokine axes is key for NK cells to contribute to organ homeostasis and the disruption of a proper NK cell trafficking is associated with pathological conditions, including excessive inflammation and cancer.

### 1.3. Natural Killer Functions

NK cells contribute to host defense through direct killing of infected and transformed cells and through the release of cytokines utilized to orchestrate other immune subsets, as discussed below. Unlike adaptive T lymphocytes, NK-cell-mediated killing is independent from human leukocyte antigen (HLA) restriction and is regulated by a repertoire of activating and inhibitory receptors that recognize germline-encoded ligands expressed by target cells [25,26]. The process that leads to NK cell activation results from the integration between stimuli activation and inhibitory signals, and the presence of receptors with apparent redundant activity is essential to allow for self-tolerance and to guarantee fine-tuning of NK-cell cytotoxic functions [27]. In accordance with this assumption, the expression of inhibitory receptors is indispensable for NK cell education [28]. Inhibitory receptors belong to the KIRs family in humans, the killer cell lectin-like receptors (KLRs) family (i.e., CD94/NKG2A) and the leukocyte immunoglobulin-like receptors (LILRs) in both humans and mice, and the KIRs functional homolog Ly49 in mice. Inhibitory receptors recognize major histocompatibility complex (MHC) class I/MHC-I-like molecules, whose absence or deregulation in infected or transformed target cells triggers the so-called missing-self recognition by NK cells. NK cells also express various inhibitory receptors that recognize non-MHC molecules, including the co-inhibitory receptors T cell Ig and ITIM domain (TIGIT) and CD96, both binding to the poliovirus receptor CD155 [29]. Interestingly, the engagement of such non-canonical inhibitory molecules, which also include programmed cell death protein 1 (PD-1) and toll-il-1 receptor 8 (TIR-8), of ligands expressed on tumor cells, transmits shut down signals to NK cells that promote immune escape and deserve attention in cancer treatment [30,31,32,33,34]. If downregulation of the inhibitory ligands is essential for NK cells to mount a productive response, activating molecules also need to be expressed by target cells and recognized by activation receptors on NK cells. In humans, examples of NK cell activation receptors include NK gene 2D (NKG2D) and the natural cytotoxicity receptor (NCR) NKp46, which recognize ligands upregulated on target cells, and CD16, which binds antibody-opsonized targets. Importantly, multiple activation receptors must be involved to unleash NK cell effector functions, and co-activation is essential for NK cells to be effective [35,36].

## 2. NK Cells in Cancer: Mechanisms of Dysfunction

Natural killer cells were first described in the late 1960s as cells capable of killing cancer cells without restriction by HLA molecules [37]. The concept of NK cells as a tool for cancer therapy was formally proved when the infusion of NK cells was first employed in the treatment of leukemia patients, with promising efficacy [38]. Nowadays, we are conscious that NK cells are key in anti-tumor immunity, given their potent cytotoxicity against cancer cells. Therapeutic approaches that take advantage of the anti-tumor activities of NK cells, such as the adoptive transfer of genetically modified NK cells and the modulation of checkpoint molecules, turned out to be promising immunotherapeutic strategies to eliminate cancer. Unfortunately, exposure to tumor cells and to the components of the tumor microenvironment (TME) impairs NK-cell effector functions and makes them dysfunctional. Dysfunctional NK cells are defined by a limited release of effector cytokines and a decreased ability to kill malignant cells. Multiple mechanisms are implicated in such dysfunction, spanning from inhibition of recruitment to the tumor bed, activation of inhibitory processes, blowing up of activation signals, and deregulation of metabolism (Figure 1). 

### 2.1. Inhibition of Recruitment

NK cells express a heterogeneous repertoire of chemoattractant receptors that are distinct for each cell subset, and thus differentially regulate the recruitment of each population to the tumor bed. CD56^bright^ NK cells are known to express CCR2, CCR5, CCR7, and CXCR3, while CD56^dim^ NK cells specifically express CXCR1, CXCR2, CX3CR1, S1P_5_, and Chemerin Receptor 23 (ChemR23) [39]. The abundance of NK cells in tumors correlates with a good prognosis in most cancers; however, NK cell infiltration inside the tumor tissue is generally limited, thus suggesting that cancer cells engage in strategies that target chemoattraction to limit the recruitment of NK cells and promote escape from killing [40].

Several chemokine/receptor axes have been described in the regulation of NK cell trafficking to the tumor bed, and some have been reported to modulate NK cell anti-tumor functions. The expression by tumor-infiltrating NK cells of certain chemokine receptors including CCR2, CCR5, CCR7 and CX3CR1, and the abundance of their respective ligands in the TME has been correlated to enhanced NK cell tumor infiltration and improved cytotoxic response [41]. However, tumor cells could hijack NK cell trafficking mechanisms to shut down their functions. For example, high levels of CCL19 have been reported in the serum of stage IV melanoma patients, suggesting a mechanism by which NK cells are retained in the bloodstream to avoid their migration to LNs [42]. On the same line, low levels of the chemokines CXCL12, CXCL10, and CCL27 have been detected in the TME of endometrial carcinoma, suggesting a potential impairment of NK cell migration to the tumor bed in this context [43]. More recently, the interaction between CXCL12, released by hepatic stellate cells, and CXCR4, expressed by NK cells, has been associated with the induction of NK cell quiescence and increased breast cancer outgrowth [44].

Interestingly, CD56^bright^ NK cells that are generally endowed with limited cytotoxic functions and immune-regulatory properties are the most abundant NK subset in many tumors, including non-small cell lung cancer (NSCLC) and breast cancer, thus suggesting that tumor cells orchestrate attraction and select which NK cell population to call. For example, transforming growth factor beta (TGF-β) in the tumor tissue promotes the deregulation of the CX3CL1–CX3CR1 axis, thus reducing NK cell infiltration in hepatocellular carcinoma and breast cancer. More generally, TGF-β signaling favors the recruitment of CD56^bright^ NK cells at the expense of the CD56^dim^ subset and in parallel impairs NK cell activation, thus representing an interesting target to improve NK-cell-mediated therapies [45,46,47]. On the same line, increased levels of CCL19, CXCL9, and CXCL10 and decreased levels of CXCL2 in lung tumors foster the recruitment of CD16^−^ NK cells, at the expense of the more cytotoxic CD16^+^ NK cells [48,49].

Interestingly, if it is generally accepted that NK cells circulate and patrol the body to search for danger, it is also established that NK cells reside in peripheral tissues and that distinct subsets show preferential organ localization [50]. Such organ-specific homing should be taken in consideration, as distinct tumors and metastatic lesions may respond differently to therapies aimed at improving NK cell recruitment. Another crucial point to consider is the spatial localization of NK cells within the tumor tissue. Indeed, evidence indicates that the abundance of NK cells in certain cancer contexts is independent from the amount and assortment of chemokines available. In addition, NK cells are often more abundant in the adjacent tissue than inside the tumor lesion [51]. Together, these data suggest that the stiffness and organization of the extracellular matrix (ECM) may impact the capability of NK cells to cross the stromal barrier around cancer [52,53]. More investigation is needed to clarify the role played by ECM on NK cells’ infiltration across cancers and may provide strategies to increase NK cell abundance in the tumor bed. 

### 2.2. Modulation by Soluble Factors

Once in the tumor, infiltrating NK cells are exposed to a variety of strategies engaged by cancer cells and components of TME to establish immunosuppression. TGF-β is one of the most abundant cytokines in the TME in various cancers, and is widely known to dampen immune surveillance. TGF-β recognition by NK cells results in the inhibition of cytokine secretion and granules release, and in the modulation of cell metabolism through the mammalian Target of Rapamycin (mTOR) pathway [54]. In addition, TGF-β produced by cancer cells and by a component of the TME has been reported to inhibit NK cell anti-tumor activity via downregulation of the activating receptor including NKG2D, NKp30, and NKp44 [55,56]. Among other soluble factors, Prostaglandin E2 (PGE_2_) has been reported to impair the recruitment and activation of NK cells in preclinical models. Genetic deletion of EP2 and EP4 receptors for PGE_2_ facilitates the early intra-tumoral accumulation of interferon gamma (IFN-γ)-producing NK cells and an IFN-γ-dependent re-education of the TME against cancer [57]. On the same line, PGE_2_ released by CAFs in hepatocellular carcinoma brings NK cells to dysfunction [58]. The influence of IL-10, a known inducer of immunosuppression that is abundant in the TME, is more complex. While in vitro results suggest that IL-10 exposure lowers the release of cytotoxic factors by NK cells and hampers activation, in vivo evidence in murine models showed that IL-10 enhances NK cell activation by itself or in combination with additional secreted factors, such as IL-18 and IL-2, and inhibits metastasis formation when infused in cancer models [59,60,61,62]. An interesting soluble factor enriched in the tumor bed is represented by adenosine, a purine nucleoside that recognizes specific receptors expressed on T cells, as well as NK cells and restrains cell activation [63]. Adenosine is produced upon the degradation of adenosine triphosphate (ATP) by ectonucleotidase CD38, CD39, and CD73, expressed by cancer cells and by components of the immune microenvironment, including T cells and macrophages. The investigation of the crosstalk between CD39-expressing immune subsets and NK cells promises new discoveries that may further clarify the mechanisms of NK cell dysfunction in cancer. Importantly, the variety and abundance of soluble molecules in the tumor is influenced by the cellular composition of TME, as most factors are released by both tumor cells and stromal subsets. The investigation of the crosstalk between NK cells and components of the TME is essential in order to distinguish the mechanisms that underlie NK cell dysfunction in cancers and is discussed below.

### 2.3. Engagement of Checkpoint Inhibitors

Similar to T lymphocytes, NK cells express a variety of checkpoint molecules that, when engaged by ligands present within the tumor, hinder cell activation and killing capability. NK cells express HLA-specific inhibitory receptors, such as KIRs, NKG2A, and Lymphocyte Activation Gene-3 (LAG-3) and non-HLA-class I-specific inhibitory receptors, including PD-1, T cell Immunoglobulin and Mucin-domain-containing molecule 3 (TIM-3), TIGIT, cluster of differentiation 112 receptor (CD112R), and CD96 [64]. PD-1 is expressed by NK cells in chronic infections, and is indicative of an exhausted state in tumor-infiltrating NK cells. Accordingly, PD-1 expression by NK cells is augmented in multiple cancers, including lymphomas, tumors of the digestive tract, ovarian cancer, breast cancer, renal adenocarcinoma, and others [65]. Interestingly, the recognition of PD-1 on NK cells by its ligands suppresses functional activation in cancer models [32]. Accordingly, the transfer of NK cells pre-incubated with a PD-1-blocking antibody sustains the recognition and killing of cancer stem cells and restrains tumor growth in a glioma model [66]. The immune cell profiling of peripheral blood in cancer patients administered with anti-PD1 immunotherapy is ongoing and will explore the impact of PD-1 engagement on NK cells, thus providing essential information on the matter (NCT02535247 and NCT01714739). Interestingly, combinatorial therapy of anti-PD-L1 and NK cell activating cytokines significantly augments the activity of NK cells against PD-L1 negative leukemia cells, thus potentially explaining the clinical response of PD-L1 negative cancers to PD-L1 inhibition [32,67]. Among other checkpoint molecules, Tim-3 has been found to be upregulated on the surface of NK cells from cancer patients affected by gastric cancer, lung cancer, renal cancer, head and neck cancer, melanoma, and multiple myeloma, among others. In murine models, Tim-3 defines exhausted NK cells with impaired cytotoxic capability, and the inhibition of Tim-3 in vitro augments effector functions and cancer killing [68,69,70]. Similar to PD-1 and Tim-3, additional HLA-independent checkpoints including CD96, CD112R, and TIGIT are increased in expression on NK cells in cancer patients and have been reported to restrain NK cell killing ability in vitro and in vivo. Interestingly, TIR8, a receptor that belongs to the IL-1 family, has recently emerged as a novel NK cell immune checkpoint in cancer and further investigation is ongoing to verify whether genetic manipulation of this receptor may be exploited for cancer therapy. In general, there is currently growing interest in the manipulation of these molecules with the aim of augmenting NK cell anti-tumor properties and improving the efficiency of NK-cell-based therapies in cancer. It has, however, to be considered that most immune checkpoints are expressed by NK cells in physiology and are considered to be involved in immunotolerance, thus raising concerns on the safety of healthy tissues upon the administration of checkpoint inhibitors. Further investigation is needed on this line to address the emerging problem of excessive immune reaction and life-threatening side effects associated with the use of immunotherapies. 

### 2.4. Impact of Hypoxia and Acidification of the TME

Hypoxia is a hallmark of solid tumors and tumor-infiltrating cells undergo hypoxic stress that sustains an immunosuppressive microenvironment. When exposed to hypoxia, tumor cells as well as stromal and immune components of the TME upregulate hypoxia inducible factor 1 subunit alpha (HIF-1α), a transcription factor that in turn modulates cell metabolism, differentiation, and activation. A first report on hypoxic stress in NK cells showed that hypoxia induces an impairment of NK cell effector functions in multiple myeloma [71]. On the same line, tumor-infiltrating NK cells have been described to upregulate Hif-1α in different tumor models and the genetic deletion of Hif-1α confers to NK cells superior effector functions and a more potent tumor killing [72]. Interestingly, the constitutive expression of a high affinity CD16 receptor and internal IL-2 in NK cells makes them resistant to the deregulation of effector functions induced by hypoxia. This evidence provides additional information on the characteristics of high affinity NK cells, currently under investigation in the clinic [73]. In liver cancer patients, exposure to the hypoxic microenvironment induces an abnormal fragmentation of the mitochondria. In this context, the mechanistic target of rapamycin-GTPase dynamin-related protein 1 (mTOR-Drp1) activation caused by low oxygen availability alters the mitochondrial programming, causes a reduction in mitochondrial mass and mitochondrial membrane potential and leads NK cells to dysfunction [74]. On the same line, exposure to lactic acid produced by colorectal cancer cells induces reactive oxygen species (ROS) accumulation and mitochondrial damage in tumor-infiltrating NK cells that provokes cell dysfunction [75]. Similarly, the upregulation of lactate dehydrogenase observed in melanoma sustains the acidification of TME with the consequent downregulation of nuclear factor of activated T cells (NFAT) in T and NK cells and lower IFN-γ release [76]. Many findings have described the abundance of lactic acid as a central regulator of immunotolerance in cancer, and modulation of the tumor acidification represents a valuable approach to improve NK and T-cell-based immunotherapies. It has, however, to be considered that both hypoxia and acidification of the microenvironment drive the upregulation of checkpoint molecules, such as PD-1 and PD-L1, on cancer cells and tumor-infiltrating immune subsets [77,78]. As a consequence, low oxygen and high lactic acid may render the tumor more responsive to immunotherapies based on checkpoint inhibitors, thus paradoxically becoming potential allies for immunotherapy. 

## 3. Crosstalk between NK Cells and Immune Subsets within the TME

The efficacy of NK-cell-mediated tumor killing depends on a self-reinforcing anti-tumor response in which NK cells, once recruited to the tumor bed, kill cancer cells and, at the same time, foster innate and adaptive immunity. During the initial steps of cell transformation, the removal of NK-cell-sensitive tumor variants leads to an “edited” pool of tumor cells. Within these cells, resistant clones may emerge, thus evolving into a primary tumor that escapes NK cell immunosurveillance through different mechanisms. In this context, immune cells, stromal cells, extracellular matrix, and regulatory proteins within the TME contribute to tumor cell proliferation, migration, and dissemination as well as to tumor escape from NK-cell-mediated killing. The crosstalk between NK cells and components of TME offers a plethora of mechanisms and molecules that may be exploited to improve NK-cell-based anti-cancer therapies (Figure 2).

### 3.1. NK Cells and Conventional T Lymphocytes 

Several works highlight the association between favorable cancer outcome and T/NK cells crosstalk [79,80], suggesting that NK cells and T lymphocytes cooperate in the establishment of an effective anti-tumor response. Indeed, NK-cell-derived IFN-γ have been reported to promote an anti-tumor T helper cell type 1 (Th1) polarization in CD4^+^ T cells [81]. Similarly, activated NK cells stimulate CD4^+^ T cell proliferation through OX40/OX40L interactions [82]. Notably, NK cells also modulate T cell response through indirect mechanisms, involving dendritic cell (DC)-mediated regulation of T cell differentiation [83,84,85,86,87]. Finally, NK cells are capable of inducing a type 17 polarization, marked by an IFN-γ- and IL-17A-producing ability, in CD8^+^ T cells via inflammatory DC priming [88]. NK cell-mediated elimination of target cells not only affects tumor cell pool, but boosts a T cell anti-tumor response. Indeed, tumor antigens released upon target apoptosis, if phagocytosed and processed by DCs, increase antigen presentation to T cells, and, as a consequence, their functionality [89]. Concomitantly, T cells impact on NK cell functionality trough the release of cytokines, including IL-2 and IL-15, that sustain NK cell survival and activation [90]. More recently, it has been demonstrated the regulation of NK cell anti-tumor response by MR1-restricted mucosal-associated invariant T (MAIT) cells. Notably, under steady state conditions MAIT cells inhibit both NK cell maturation and anti-tumor response, however, once activated, MAIT cells enhanced NK cell activation and anti-tumor immunity in an IFN-γ-dependent manner in melanoma and breast cancer mouse tumor models [91].

All of the above-mentioned interactions between NK cells and T cells represent promising therapeutic targets in cancer, and several works testify to the advantages of this strategy. Indeed, the synergistic activation of both T and NK cell anti-tumor response, via MHC class I chain-related protein A/B (MICA/MICB) stress molecules, Stimulator of Interferon Genes (STING) agonists, or IL-2 superkines, were found to be highly effective in controlling tumor growth, especially in MHC-I-deficient tumors and in poorly immune infiltrated microenvironments [80,92]. Similarly, harnessing the MAIT cell capability to promote NK-cell-mediated anti-tumor response represents a sustainable therapy for established tumors [91].

### 3.2. NK Cells and T Regulatory Cells

Increased frequency of T regulatory cells (Treg) is a common trait of several cancer types, and is positively correlated with cancer progression [93,94]. Notably, Treg cell abundance has been negatively correlated with NK cell frequency and functional activation, thus suggesting a Treg-specific suppression of NK cell anti-tumor immunity [95,96,97]. Several works agree on TGF-β as a key molecule involved in Treg-driven NK cell inhibition because of its ability to suppress NK-cell-mediated tumor killing and to reduce NKG2D and IFN-γ expression [95,98,99,100,101,102]. Accordingly, the direct blockade of the suppressive cytokines TGF-β and IL-10 by means of inhibitory monoclonal antibodies was shown to be effective at improving NK cell recognition of target cells and re-activating the NK cell cytotoxic ability [103,104]. Similar to TGF-β, the anti-inflammatory cytokine IL-10, released by activated Treg cells, has been reported to modulate the expression of NKG2D ligands, including MIC and UL16-binding proteins (ULBP), thus determining an impairment of NKG2D-mediated lysis of the target cells [103]. An additional immunosuppressive mechanism is the accumulation of extracellular adenosine (ADO) within the tumor microenvironment, a direct consequence of hypoxia-driven purinergic signaling. This metabolite, produced by both cancer cells and Treg cells via CD39 and CD73 enzymes, contributes to tumor development, by negatively regulating the trafficking, proliferation, and functionality of NK cells [105,106,107,108]. Finally, the competition for IL-2 within the microenvironment is an additional strategy exploited by Treg cells to mediate NK cell suppression. As IL-2 improves the ability of NK cells to engage target cells, the high consumption of this molecule by Treg cells may restrain NK cell cytotoxicity [109]. Importantly, Treg cells could directly block NK cell anti-tumor activity in vivo by killing NK cells in a granzyme-B-dependent manner [110]. In accordance with this evidence, strategies aimed at depleting Treg cells have resulted in the concomitant restoration of NK cell functionality, including signal transducer and activator of transcription 3 (STAT3) modulation, cytotoxic T-lymphocyte antigen 4 (CTLA-4) inhibition, low oral doses of cyclophosphamide [111], anti-OX40 monoclonal antibody [112], and anti-cancer agents such as Lenalidomide and Pomalidomide [113,114]. In addition, the combination targeting of Treg and NK cells in cancer has been proposed and represents a promising therapeutic strategy. Among others, the administration of an IL-2/anti-IL-2 monoclonal antibody (mAb) complex that allows for the specific conveyance of cytokines to NK cells [115] has been described. Similarly, the activation of the IL-2Rγ chain signaling via IL-2, IL-4, IL-7, and IL-12 overcomes the inhibitory effect of Treg cells on NK cell lysis [95]. Finally, the combinatorial strategy based on the anti-PD-1 antibody Nivolumab together with the Treg-depleting antibody Mogamulizumab was shown to be a safe and potentially effective option in advanced and metastatic solid tumors. Indeed, Mogamulizumab leads to NK-cell-mediated CCR4^+^ Treg cell depletion, which in turn contributes to the activation of CD8^+^ T cell and NK cell anti-tumor response [116]. 

### 3.3. NK Cells and DCs

Several works have explored the complex crosstalk existing between NK cells and DCs within the tumor microenvironment. Importantly, the encounter between these two cell subsets favors their bidirectional activation, maturation, and functional activity, collectively inducing a potent anti-tumor response [117]. Because of the low expression of MHC-I molecules on immature DCs, NK cells are capable of specifically killing them, in parallel sparing properly activated DCs, thus favoring the activation of an anti-tumor immune response [118]. NK cells not only select immunogenic DCs, but they also favor their maturation via IFN-γ, tumor necrosis factor alpha (TNF-α), and TNF superfamily member 14 (TNFSF14) in a dependent manner [119]. NK cells also impact on DC recruitment to the tumor bed through the release of a panel of chemokines, including CCL5, X-C Motif Chemokine Ligand 1 (XCL1), and XCL2 [120]. On the other side, tumor cells counteract this axis by releasing PGE_2_, a molecule that concomitantly inhibits NK cell functionality and DC recruitment via X-C Motif Chemokine Receptor 1 (XCR1) and CCR5 downregulation [57]. In contrast, engagement of PD-1 and CTLA-4 on NK cells has been reported to negatively regulate DC maturation and to provoke a defective CD8^+^ T cell priming within the microenvironment in murine fibrosarcoma and lung cancer [121,122].

In parallel, DCs are able to sustain NK cells through the release of inflammatory cytokines. In particular, IL-15, IL-12, and IL-18 secreted by DCs are strong inducers of NK cell proliferation, survival, and anti-tumor response [123,124,125], and are critically involved in the regulation of the anti-metastatic potential of NK cells.

NK cells, as previously mentioned, could act as a bridge between DC/T cells, determining the type of polarization acquired by DCs and, as a consequence, by T cell response [83,84,85,86,87,88]. 

The positive feedback loop existing between NK cells and DCs could be considered a promising target for enhancing the anti-tumor response. Preclinical tumor models have demonstrated that the abrogation of the CD47/signal regulatory protein alpha (SIRPα) axis is sufficient to trigger DC/NK cell crosstalk. Indeed, the administration of an anti-CD47 mAb was able to significantly improve the uptake of tumor-derived DNA by DCs, thus determining the activation of the cGAS/STING pathway and, as a consequence, the infiltration and activation of NK cells via CXCL9 and IL-12 [126]. Similarly, the adoptive transfer of DCs and NK cells together with a suboptimal dose of doxorubicin determine a significant therapeutic efficacy, as well as the enhanced expression of co-stimulatory molecules in DCs and the suppression of Treg cell function [127]. Importantly, the majority of clinical trials using DC-based cancer vaccines highlight how the patient outcome is positively correlated with NK cell frequency and functionality, thus further supporting that DC activation by DC-targeting strategies promotes not only DC antigen-presenting function, but also the NK cell killing ability [128]. In parallel, therapeutic strategies aimed at improving NK functionality not only determine an increase in lytic ability, but they additionally enhance anti-tumor adaptive immunity via the selection of immunogenic DCs [129].

### 3.4. NK Cells and Neutrophils

Neutrophils are a highly heterogenous cell subset, endowed with both pro-tumoral and anti-tumoral functions, on the basis of the tumor type and stage of progression. Accordingly, within the tumor microenvironment, neutrophils can exert both an activation and an inhibitory effect on NK cell functionality via both soluble mediators and cell-to-cell interaction mechanisms [130]. PD-L1^+^ neutrophils have been reported to cause the downregulation of the chemokine receptor CCR1, thus limiting the recruitment of tumoral NK cells and reducing the responsiveness of NK cell-activating receptor NKp46 and NKG2D, critically involved in stress ligand recognition and the subsequent elimination of transformed cells [131]. The negative modulation of the activating receptor NKp46 is one of the main inhibitory mechanisms exploited by tumor-associated neutrophils. Indeed, both Cathepsin G and reactive oxygen species (ROS) released by neutrophils negatively affect NKp46 expression, specifically within the cytotoxic CD56^dim^CD16^+^ NK cell subset [132,133]. On the same line, neutrophil release of Arginase I (ARG1), and the consequent arginine depletion within the tumor environment, was able to potently abrogate NK cell proliferation as well as NK cell-mediated IFN-γ release [134]. Finally, neutrophil extracellular traps (NETs) generated by neutrophils are able to physically inhibit CD8^+^ T cell- and NK cell-mediated cytotoxicity by impairing the effector/target interaction [135]. As mentioned before, depending on the conditions, neutrophils may sustain NK cell functionality. Neutrophil-derived elastase and lactoferrin contribute to the reinforcement of the NK cell cytotoxic ability [136] and, similarly, neutrophil-derived IL-12 improves NK cell functionality by inducing IFN-γ and perforin production [137]. Interestingly, NK cells have been described to modulate neutrophil functions. In particular, IFN-γ released by NK cells is able to abrogate the angiogenic activity of tumor-associated neutrophils by suppressing vascular endothelial growth factor A (VEGF-A), thus restraining tumor growth [138]. From a therapeutic point of view, some works have reported that the blockade of the PD-1/PD-L1 axis by means of neutralizing antibodies antagonizes the inhibitory effects of neutrophils on NK cells [131]. On the same line, the use of Cathepsin G inhibitors counteracts the decrease in NKp46 expression in NK cells upon neutrophil exposure [132]. Finally, interfering with NETs could improve NK cell recognition of target cells and their subsequent elimination [135].

### 3.5. NK Cells and Macrophages

Similar to neutrophils, tumor associated macrophages (TAMs) exhibit a continuum phenotype between the M1- and M2-like status within the TME [139]. As a consequence, the interaction between macrophages and NK cells may result in very different outcomes. Several pro-inflammatory cytokines released by M1-activated macrophages, including IL-12, IL-15, IL-18, and TNF-α, are critically involved in the improvement of NK cell anti-tumor function [140]. Notably, macrophage exposure to damage-associated molecular patterns (DAMPs) and tumor-associated molecular patterns (TAMPs) within the TME is a key factor able to trigger M1-polarization and the release of NK cell-activating cytokines [141,142,143]. Importantly, the expression of the above-mentioned molecules has been associated with a favorable outcome in cancer patients [144], possibly because of their ability to reduce the NK cell activation threshold [34,145]. On the same line, Membrane Spanning 4-Domains A4A (MS4A4A)^+^ macrophages have been described for their anti-tumor function, in view of their ability to release pro-inflammatory cytokines, including IL-15 and IL-18, upon engagement of Dectin-1 by tumor cells. Notably, these molecules promote NK cell-mediated cytotoxicity, IFN-γ, and resistance to metastasis [143]. On the contrary, very recent evidence indicates that triggering receptor expressed on myeloid cells 2 (TREM2) confers a pro-tumorigenic program to macrophages that interferes with IL-15 release by tumor-infiltrating DCs, thus negatively affecting NK cell activity [146].

Other cell-to-cell contact-mediated interactions contribute to the reinforcing feedback loop able to polarize macrophages towards an anti-tumor function and to improve NK cell cytotoxicity. In particular, CD48/2B4 and NKG2DL/NKG2D axes both contribute to the positive bi-directional regulation of the anti-tumor immune response [147,148,149]. On the contrary, alternatively activated M2-like macrophages exert a strong inhibitory effect on NK cells by releasing potent inhibitory factors, such as TGF-β and IL-10. On the other side, NK cells can modulate macrophage dynamics within the TME by inducing a switch towards either a pro-tumor or an anti-tumor phenotype. In particular, once activated, NK cells release cytokines, including IFN-γ, TNF-α, and granulocyte-macrophage colony-stimulating factor (GM-CSF), which induce in macrophages an M1-like polarization and secretion of IL-12 and IL-18 [150,151,152]. In a different setting, tumor-associated NK cells have been reported to promote M2-like polarization via IL-10 release [153]. 

The macrophage–NK cell crosstalk in cancer may be targeted at different levels, thus offering multiple therapeutic options. A first strategy could be the blockade of TGF-β signaling, which is known to have a potent inhibitory effect on NK cell functionality. Importantly, TGF-β inhibits IL-15-dependent NK cell proliferation and activation, while it does not activate any pro-apoptotic role in NK cells. Therefore, its therapeutic targeting will allow for the almost complete restoration of NK cell-mediated anti-tumor immunity [154]. On the same line, blocking macrophage receptor with collagenous structure (MARCO) receptor on tumor-associated macrophages resulted in TAM repolarization as well as in an improvement in NK cell anti-tumor response both in primary and metastatic tumors. Importantly, the combinatorial blockade of MARCO and PD-1/PD-L1 significantly improve anti-tumor immunity, thus suggesting a new possible therapeutic strategy [155]. 

### 3.6. NK Cells and Monocytes

The role of monocytes in the regulation of NK cell functionality within the TME has been poorly described; however, some works highlight the impact of monocytes in a metastatic context. In particular, the so-called patrolling monocytes, endowed with the ability to quickly sense early metastasis and to activate an anti-tumor immunity, exert a crucial role in the recruitment and activation of NK cells at metastatic sites. In particular, patrolling monocytes release IL-15 in response to primary tumors, thus inducing NK cell anti-metastatic response. Notably, these cells highly express effector molecules, including IFN-γ, perforin, and granzyme, while in the absence of patrolling monocytes, NK cells were more inhibited, as demonstrated by the impaired expression of the activating receptor Ly49D and increased expression of the inhibitory receptor NKG2A/CD94 [156,157,158].

## 4. NK Cells as Target for Anti-Cancer Therapies

NK cells are endowed with tumor killing capabilities and thus represent a valuable asset for anti-cancer therapies. The ideal therapeutic approach should evade those strategies that are engaged by cancer cells to hinder NK cell functionality, and in parallel improve NK cell recruitment and activation [39]. The adoptive transfer of autologous or allogeneic NK cells has proven to be efficient in liquid cancers, and multiple trials are currently investigating the efficacy of such an approach in solid tumors. However, if the infusion of NK cells increases the abundance of NK cells in the tumor-bearing host, cells that reach the tumor are still exposed to strategies of evasion that limit efficiency. This is the reason the modification of infused NK cells is necessary to sustain their function. Engineering strategies include the overexpression of activating factors, the downregulation of inhibitory molecules, and the augmentation of specificity. Among the latter, genetic engineering to insert chimeric antigen receptors (CARs) represents a turning point in NK-cell-based therapies [159,160]. CARs are synthetic fusion proteins equipped with an extracellular domain that recognizes the antigen and an intracellular portion that triggers cell activation. Multiple CAR-NK cells have been produced, and extensive research in the field of CAR-T lymphocytes has generated knowledge to refine receptor activity and specificity that can be easily applied to NK cells. A strong efficacy has been reached with the combination of cytokine exposure and genetic engineering to insert CARs. Armoring NK cells with cytokines before infusion confers them a superior killing ability and longer persistence. Interestingly, NK cells cultured in vitro with IL-12, IL-15, and IL-18 acquire a memory-like phenotype associated with stronger effector functions and longer persistence in vivo [12,161]. On the same line, the genetic modification of NK cells that results in autocrine cytokine synthesis showed promising therapeutic efficacy in patients with lymphoid malignancies, including CD19*^+^* hematological tumors [159,162]. It has to be said that the genetic modification of NK cells implies a considerable effort, and alternative strategies are under evaluation. Among these, the use of Bispecific and Trispecific Killer cell Engagers (BIKEs and TriKEs) that sustain activation receptors and in parallel direct NK cells to the target holds strong promise. Killer engagers combine the Fv domains binding CD16 or other activating receptors, including NKG2D, Nkp30, Nkp44, and Nkp46, with Fv domains that recognize single or dual tumor antigens. Phase I and I/II clinical trials testing BIKEs and TriKEs are currently underway for hematological malignancies and solid tumors, and have shown promising preliminary results [163]. One of the major obstacles is represented by the fact that the development of engagers implies knowledge of tumor antigens. Furthermore, the extent of NK cell functionality upon the recognition of the ligands by the engagers in vivo is partially unknown, and whether excessive or uncontrolled cell activation may lead to dysfunction still needs to be investigated. The strategies listed above are based on the improvement of NK cell activation by different means. A diverse strategy currently under exploration points, instead, at disentangling inhibitory signals. NK cells are known to express immune checkpoint molecules, including PD-1, TIM-3, and TIGIT, that once engaged, may limit their effector functions [64,164]. Disruption of such checkpoints by mean of neutralizing antibodies showed efficacy in preclinical models and was reported to be partially dependent on NK cells, as discussed below. Among other evidence, PD-1 and PDL-1 blockade has been reported to be dependent on NK cells in multiple pre-clinical models of cancer, including, lymphoma and prostate cancer [32]. Both TIM-3 and TIGIT are upregulated in tumor-infiltrating NK cells. In both melanoma models and patients, TIM-3 and TIGIT define subsets of NK cells showing an exhausted phenotype and impaired cytotoxic functions; accordingly, TIM-3 and TIGIT inhibition reverses NK cell exhaustion and dysfunction [69,165,166]. It has to be said that immune checkpoints have been deeply investigated in T lymphocytes, while their role in NK cells is still under debate, and further investigation is needed to dissect their potential as immunotherapy targets.

## 5. NK and T Cell-Targeting Immunotherapies

Preclinical studies have demonstrated the impact of immune checkpoint blockade (ICB) on NK cell functionality. In particular, PD-1 has been reported to be a key checkpoint for NK cell activation in several mouse cancer models. Accordingly, the efficacy of the PD-1/PD-L1 blockade relies on both T cells and NK cells [32]. Similarly, the CTLA-4 inhibitor antibody Ipilimumab was shown to be effective in inducing IL-2 expression as well as in improving NK cell cytotoxicity in a murine model of melanoma, even if the expression of CTLA-4 on NK cells remains controversial [167]. Notably, recent data collected from clinical trials have demonstrated the positive impact of ICB on NK cell reactivation in tumor patients. In particular, the anti-PD-1 blocking antibody Nivolumab has been approved for the treatment of relapsing or refractory Hodgkin’s lymphoma after hematopoietic stem cell transplantation. Importantly, in this context, treatment with Nivolumab was able to accelerate the expansion of NK cells upon transplantation and make them more functional, thus limiting possible tumor growth [168]. On the same line, PD-1 blockade by means of Nivolumab was shown to be effective at restoring NK cell activation in combination with the anti-endothelial growth factor receptor (EGFR) inhibitor Cetuximab. Notably, Cetuximab promote the expression of both PD-1 on NK cells and PD-L1 on tumor cells. However, the establishment of the inhibitory interactions between NK and target cells allow the PD-1 blockade to be effective [169]. In addition, some works highlight the significant enhancement of CAR-T cell anti-tumor activity in the presence of anti-PD1 blockade treatment, as demonstrated by both preclinical and clinical studies [170,171,172].

Collectively, all these data support the hypothesis of a combinatorial approach based on NK cell adoptive transfer and ICB treatment to further improve the outcome of tumor patients, especially for patients with solid tumors. Importantly, there are three main options to inhibit immune checkpoint (IC) molecules, including systemic ICB therapy, CAR-NK cells expressing ICB molecules, or CAR-NK cells genetically knockout for checkpoint molecules. To date, few works have explored this strategy and have obtained encouraging results. Recently, Oyer et al., in a preclinical setting, demonstrated that the adoptive transfer of ex vivo expanded NK cells determines the upregulation of PD-L1 in tumor cells via IFN-γ release and, as a consequence, the expansion of the Treg compartment. Notably, in view of PD-L1 expression on the target cells, the anti-PD-L1 treatment was found to be highly effective, not only at inducing tumor shrinkage, but also improving NK cell survival, supporting the expansion of the cytotoxic CD57^+^ subset, and determining the re-expression of CD16. Additionally, PD-L1 blockade could counteract Treg cell induction and, as consequence, determine an improvement in adoptive NK cell therapy efficacy [173]. On the same line, ex vivo IL-2-activated NK cells were shown to be ineffective at inhibiting gastric cancer growth as a monotherapy; however, the combination with the anti-PD-1 antibody Nivolumab, administered as systemic therapy or as a pretreatment before NK cell infusion, greatly improved NK-cell-mediated anti-tumor response [174]. Apart from direct NK cell activation, ICB could also have a positive impact on the tumor infiltrate, as demonstrated by Shevtsov et al. in different preclinical models. Indeed, the infusion ex vivo 70-kDa heat shock protein (Hsp70)/IL-2-activated NK cells together with the anti-PD-1 antibody was shown to be highly effective in local tumor control as well as improving the infiltration of CD8^+^ cytotoxic lymphocytes and NK1.1^+^ cells in the tumor bed [175]. A similar approach reached the clinic and showed promising results in terms of long-term tumor control in one patient with advanced NSCLC [176]. More recently, Strassheimer et al. demonstrated that the combined treatment with human epidermal growth factor receptor 2 (HER2)-specific NK-92/5.28.z cells and anti-PD-1 blockade had a strong positive impact on tumor regression, long-term survival, and on cytotoxic lymphocyte infiltration [177]. In addition, PD-L1 CAR-NK cells in combination with the anti-PD-1 blockade antibody Nivolumab showed a synergistic anti-tumor response in a humanized mouse nasopharyngeal carcinoma patient-derived xenograft model [178]. Finally, the combination of anti-PD-L1 blocking antibody Atezolizumab in combination with the anti-Prostate Specific Membrane Antigen (PSMA) CAR NK-92 cells was shown to be highly effective in the control of castration-resistant prostate cancer (CRPC). The concomitant inhibition of the PD-L1/PD-1 axis and the direct activation of PD-L1-expressing CAR NK-92 provided a strong anti-tumor effect compared with CAR NK-92 cells alone [179].

Recently, the results of the first clinical trial (NCT02843204) aimed at evaluating the safety and efficacy of the combination of Pembrolizumab and NK cell infusion were published; 109 patients with advanced NSCLC were enrolled in the study and the impact of combinatorial therapy was tested in comparison with Pembrolizumab as a monotherapy. Importantly, patients receiving the combination therapy had a higher overall survival (15.5 months vs. 13.3 months) and a higher progression-free survival (6.5 months vs. 4.3 months). Notably, patients who received multiple NK cell infusions had a better overall survival (18.5 months) compared with those who received a single NK cell infusion (13.5 months) [180]. On the same line, a phase II study (NCT04847466) is currently testing the efficacy of the combination of irradiated PD-L1 targeting high-affinity (t-ha) NK cells with anti-PD-1 blocking antibody Pembrolizumab and the immunostimulatory molecule N-803 in patients with advanced forms of gastric or head and neck cancer (https://clinicaltrials.gov/ct2/show/NCT04847466, accessed on 29 May 2023). Additionally, PD-L1 t-haNK cells are currently being used in a phase IIb clinical trial (NCT03228667) for the treatment of patients with solid tumors that have progressed and/or relapsed after PD-1/PD-L1 checkpoint inhibitor (https://clinicaltrials.gov/ct2/show/results/NCT03228667, accessed on 29 May 2023). Finally, the safety and efficacy of FT500, an iPSC-derived NK cell product, in combination with Nivolumab, Pembrolizumab, or Atezolizumab, are being evaluated in a phase I clinical trial (https://clinicaltrials.gov/ct2/show/NCT03841110, accessed on 29 May 2023).

## 6. Conclusions

Immunotherapies represent a turning point in cancer treatment. In this context, cell-based approaches have shown relevant efficacy in many tumors. However, the employment of T cell infusions in cancer patients are limited by MHC- and antigen-restriction, excessive immune response, and autologous restrainment. NK cells embody a new opportunity that overcomes such limitations and is thus endowed with a promising therapeutic potential. However, if NK-cell-based therapies are effective against hematological malignancies, their clinical impact is still insufficient in solid tumors. Consequently, the exploration of the mechanisms underneath NK cell recruitment, metabolism, and activation will be key to unveil the causes of dysfunction and to identify novel approaches that improve NK-cell-based therapies against cancer.

## Figures and Tables

**Figure 1 ijms-24-09521-f001:**
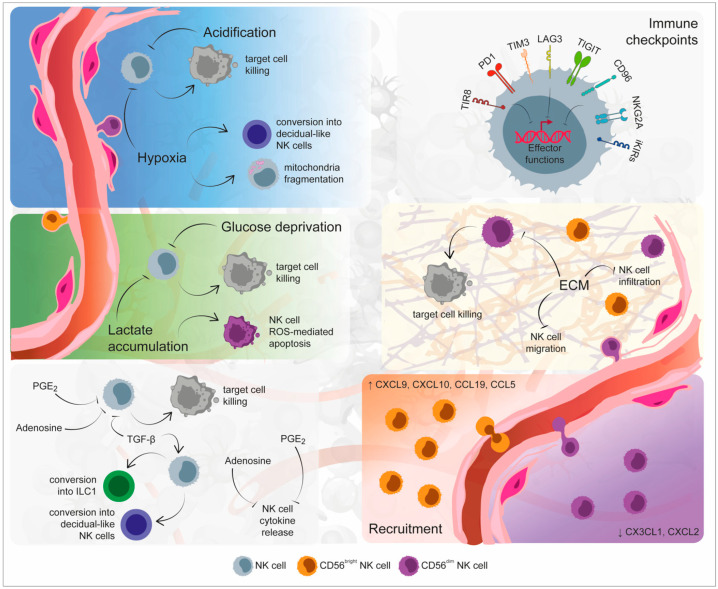
Mechanisms of Natural Killer (NK) cell dysfunction within the tumor microenvironment. The specific environmental conditions that mark the tumor site are responsible for NK cell exhaustion and dysfunction. Indeed, hypoxia, low pH, glucose deprivation, and lactate accumulation could determine both the inhibition of NK cell cytolytic activity and NK cell apoptosis. Soluble factors released by immune, tumor, and stromal cells, including transforming growth factor (TGF)-β, adenosine, and prostaglandin E_2_ (PGE_2_), not only inhibit NK cell functionality, but they additionally determine their conversion toward a non-cytolytic phenotype. An additional inhibitory mechanism exploited by the tumor microenvironment is represented by the imbalance in immune checkpoint molecules, responsible for NK cell exhausted status. The recruitment and infiltration of NK cells within the tumor site are inhibited by the alteration of extracellular matrix stiffness as well as by the imbalanced presence of chemokines. Created partly with BioRender.com.

**Figure 2 ijms-24-09521-f002:**
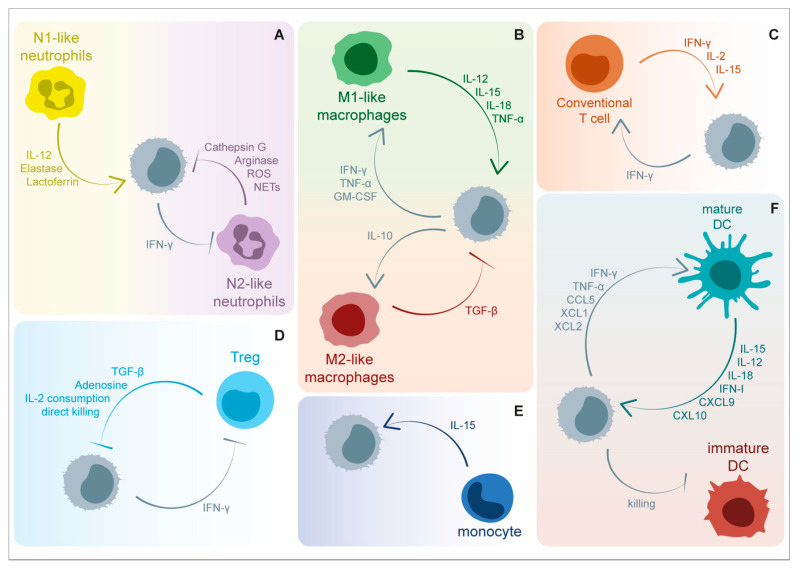
Soluble factors involved in the bi-directional crosstalk between NK cells and immune cells within tumor microenvironment (TME). NK-cell-mediated anti-tumor functions are positively and negatively modulated by a great variety of cytokines and other soluble factors released by immune cells. (**A**) Neutrophils are able to negatively modulate the expression of NKp46 through the release of Cathepsin G and reactive oxygen species (ROS). Additionally, arginine depletion and neutrophil extracellular trap (NET) release are involved in the blockade of NK-cell effector functions, including IFN-γ release and target cell killing. On the contrary, neutrophil-derived elastase, lactoferrin, and interleukin (IL)-12 enhance NK-cell functionality. NK-cell-released IFN-γ suppresses angiogenic activity of tumor-associated neutrophils. (**B**) M1-activated tumor associated macrophages are involved in NK cell triggering via IL-12, IL-15, IL-18, and tumor necrosis factor (TNF)-α, while M2-polarized macrophages counteract the above-mentioned activation mechanisms through transforming growth factor (TGF)-β release. On the other side, NK cells can foster macrophage functions through the secretion of IFN-γ, TNF-α, granulocyte-macrophage colony-stimulating factor (GM-CSF), and IL-10. (**C**) NK-cell-derived IFN-γ promotes anti-tumor T helper cell type 1 (Th1) polarization in CD4^+^ T cells and the cytolytic functions of CD8^+^ T cells. Concomitantly, T cells release IL-2 and IL-15, cytokines that are critically involved in NK cell survival and activation. (**D**) NK-cell-mediated tumor killing and IFN-γ release are negatively affected by immunosuppressive molecules, including TGF-β and adenosine, produced by regulatory T cells (Treg). The competition for IL-2 consumption represents an additional strategy exploited by regulatory T cells to suppress NK cell functions together with granzyme B-mediated direct killing. (**E**) Patrolling monocytes sustain NK cell anti-metastatic functions by IL-15 release. (**F**) The interaction between NK cells and Dendritic Cells (DCs) results in reciprocal activation. Mature DCs release cytokines able to promote NK cell activation and recruitment, including IL-12, IL-15, IL-18, C-X-C Motif Chemokine Ligand 9 (CXCL9), and CXCL10. Direct killing of immature DCs further foster optimal activation of the DC-mediated immune response. On the other side, the activation and recruitment of DCs at the tumor bed is favored by IFN-γ, TNF-α, CCL5, X-C Motif Chemokine Ligand 1 (XCL1), and XCL2. Created partly with BioRender.com.

## Data Availability

No new data were created nor generated in this manuscript.

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
