# Peer review of "NK Cells in Cancer: Mechanisms of Dysfunction and Therapeutic Potential"

_ijms, 2023, doi:10.3390/ijms24119521_

Round 1
Reviewer 1 Report
This review article is well-written and provides a comprehensive exploration of the mechanisms underlying NK cell dysfunction in cancer. The graphical abstracts are not only aesthetically pleasing but also effectively convey the main concepts. However, there are a few concerns that should be addressed:
It would be beneficial to include a discussion on the various chemokine receptors involved in the organ-specific homing of NK cells, such as those relevant to bone marrow, lung, liver, and brain.
In addition to discussing the involvement of TGF-b in impairing NK cell chemotaxis, it would be valuable to address the overall chemokine milieu within solid tumors.
Section 3.3 would benefit from the inclusion of important cytokines, namely IL-15 and IL-12, which play a significant role in the regulation of NK cell functions by DCs.
The authors are encouraged to cite the relevant articles on NK cell dysfunction, specifically PMID 34077953 and 33568351, as they contribute important insights to the topic.
Reviewer 2 Report
The field of immuno-oncology has revolutionized cancer patient care. Much of the focus in the field has been on exploiting the power of the adaptive immune response through therapeutic targeting of T cells. However, also alternative strategies, such as engaging the innate immune system, have become an intense area of focus in the field. In the review „ NK cells in cancer: mechanisms of dysfunction and therapeutic potential”, the authors gives an overview on the mechanisms that restrain NK cell anti-tumor function, and on forthcoming solutions to improve efficiency of NK cell-based therapies.
The manuscript is well organized and written. After an “Introduction” into NK cell origin and functions, the review encompasses the chapters “NK cells in cancer: mechanisms of dysfunction”, “Crosstalk between NK cells and immune subsets within the tumor microenvironment”, “NK cells as target for anti-cancer therapies”, and “NK and T cell-targeting immunotherapies”. Two figures complement the text: “Mechanisms of NK cell dysfunction within the tumor microenvironment”, and “Soluble factors involved in the bi-directional crosstalk between NK cells and immune cells within the tumor microenvironment”. 160 references give a comprehensive picture.
I do not see any critical points of this manuscript. One can only congratulate both the authors to this successful review!
